# ARGenSeg: Image Segmentation with Autoregressive Image Generation Model

**Xiaolong Wang,   Lixiang Ru,   Ziyuan Huang,   Kaixiang Ji,   Dandan Zheng
Jingdong Chen,   Jun Zhou**

Ant Group
{xiaowang.wxl, rulixiang.rlx, pishi.hzy, kaixiang.jkx, yuandan.zdd}@antgroup.com
{jingdongchen.cjd, jun.zhoujun}@antgroup.com

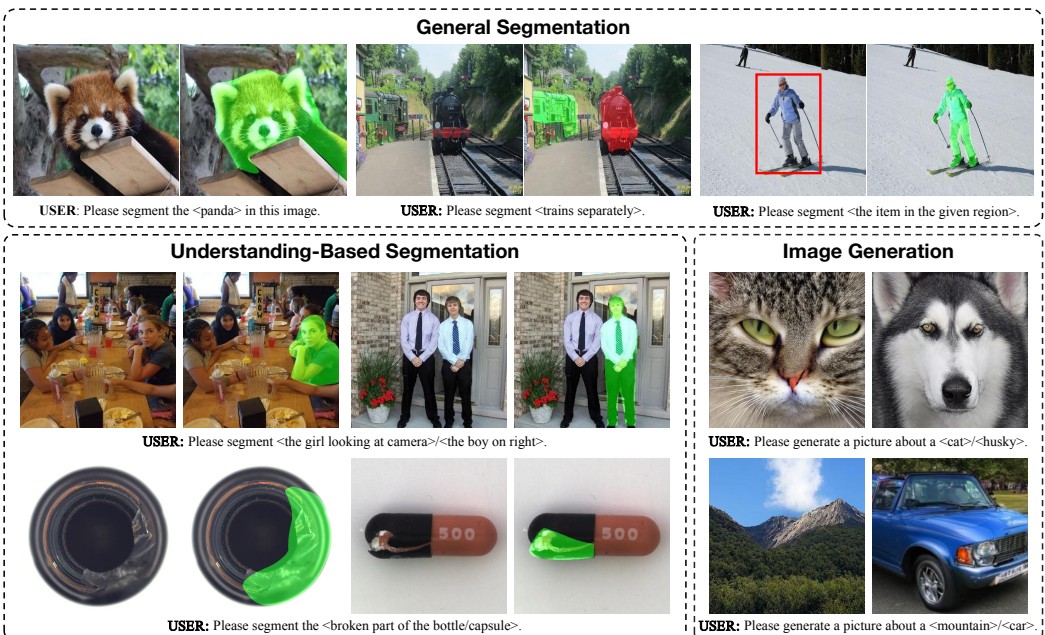

Figure 1: ARGenSeg is a unified framework for visual understanding, segmentation, and generation. It supports semantic, instance, interactive, and zero-shot reasoning segmentation, as well as anomaly detection, by leveraging strong visual understanding capabilities.

## Abstract

We propose a novel **Auto**Regressive **Gen**eration-based paradigm for image **Seg**mentation (ARGenSeg), achieving multimodal understanding and pixel-level perception within a unified framework. Prior works integrating image segmentation into multimodal large language models (MLLMs) typically employ either boundary points representation or dedicated segmentation heads. These methods rely on discrete representations or semantic prompts fed into task-specific decoders, which limits the ability of the MLLM to capture fine-grained visual details. To address these challenges, we introduce a segmentation framework for MLLM based on image generation, which naturally produces dense masks for target objects. We leverage MLLM to output visual tokens and detokenize them into images using an universal VQ-VAE, making the segmentation fully dependent

on the pixel-level understanding of the MLLM. To reduce inference latency, we employ a next-scale-prediction strategy to generate required visual tokens in parallel. Extensive experiments demonstrate that our method surpasses prior state-of-the-art approaches on multiple segmentation datasets with a remarkable boost in inference speed, while maintaining strong understanding capabilities.

# 1 Introduction

The emergence of large language models (LLMs) [8, 17, 52] has significantly accelerated the development of artificial general intelligence (AGI) [9]. Breakthroughs like ChatGPT [40] enable the transformer-based [54] autoregressive framework to unify diverse tasks of natural language processing [1, 4]. As for multimodal task, LLaVA [34] employs visual adaptor to map visual features into the embedding space of LLMs, establishing a universal paradigm for multimodal large language models [33, 5, 15, 56]. Recent studies [48, 64, 49, 67, 51, 61, 2, 3, 22] explore the unified framework for multimodal understanding and generation. However, integrating fundamental visual perception tasks into a unified AGI framework remains an open challenge. While sparse-output tasks such as visual grounding can be directly addressed via text expression [14], tasks requiring dense outpus like image segmentation are inherently difficult to represent through natural language.

Previous methods that incorporate image segmentation into MLLMs typically fall into two categories. The first discretizes dense masks into boundary point sequences [11, 42, 58], which inevitably leads to incomplete segmentation masks and unnatural object boundaries. The second achieves segmentation through downstream dedicated decoders (*e.g.*, SAM [27], Mask2Former [16]), which are conditioned on either textual prompts [12] or hidden states [29, 45, 75] generated by MLLMs. This not only results in complex model architectures, but also leads to insufficient understanding of pixel-level information for LLM due to its reliance on specialized task head.

To address the above challenges, we propose ARGenSeg, which leverages the image generation-based paradigm to integrate image segmentation into a unified MLLM framework. To retain the strong understanding capability of MLLMs, we use continuous image features as the input. For the generation output, we train the model to directly predict quantized image tokens, aligning with the next-token autoregressive prediction mechanism of language models. We use a pre-trained VQ-VAE as image tokenizer to quantize and detokenize images, with its visual tokens added to the codebook of MLLM. By leveraging the understanding ability of MLLM, ARGenSeg is capable of additional complex reasoning segmentation [29], anomaly detection [7, 6] and other image segmentation tasks [75] as shown in Fig. 1. The image tokenizer is kept frozen throughout training, thereby avoiding the dependence of LLM on subsequent decoders when learning pixel-level information.

In real-world application, image segmentation often requires fast response times. For this purpose, we adopt a next-scale prediction strategy for image generation. On one hand, the multi-scale mask generation process aligns with the intuitive process of object segmentation, which typically involves coarse localization followed by fine-grained boundary refinement. On the other hand, generating visual tokens in parallel provides a significant efficiency advantage, achieving over $4\times$ speedup compared to sequential generation methods [19, 59].

Some methods also propose to use image generation for image segmentation. UniGS [43] uses diffusion model [21, 46] to achieve image segmentation. However, its U-Net structure causes lack of understanding ability. HiMTok [57] proposes an innovative mask tokenizer that enables decoding discrete outputs from the MLLM into binary masks via image generation. However, the task-specific tokenizer limits its generality and extensibility. Moreover, both of these methods suffer from significant disadvantages in inference speed.

Extensive experiments demonstrate that the proposed ARGenSeg outperforms existing MLLM-based segmentation methods, while also achieving significantly faster inference. Notably, our method achieves superior performance using substantially less segmentation data compared to prior state-of-the-art approach [57]. In addition, the use of a general-purpose visual tokenizer provides the flexibility to extend the framework to additional tasks. As a demonstration, by fine-tuning on a small amount of image generation data, we successfully unlock the image generation capability of our framework, as illustrated in Fig. 1.

The main contributions of this paper include:

- We propose a novel image segmentation framework based on a unified multimodal understanding and generation paradigm. To our knowledge, we are the first to show that unified MLLMs can achieve SOTA segmentation results without any extra segmentation heads.

- We leverage a universal image tokenizer, allowing segmentation to fully rely on the pixel-level visual understanding of the MLLM. We further show that direct image token prediction by the MLLM is important for achieving high segmentation accuracy.

- We propose to use next-scale prediction to speed up inference. And we observe that the coarse-to-fine multi-scale mask generation process also boosts segmentation robustness.

## 2   Related Work

**Integrating image segmentation into MLLMs** not only equips them with fine-grained visual perception, but also enables more complex reasoning-based segmentation tasks by leveraging understanding capabilities. However, representing segmentation masks within the MLLM framework remains a significant challenge. PolyFormer [35] and VistaLLM [42] represent masks as polygons using point sequences, which are easy to express but struggle with complex shapes. LISA [29] aggregates segmentation information using special tokens and predicts masks through a SAM [27] decoder. Subsequent works such as GLaMM [44], PixelLM [45], GSVA [65], and PSALM [75] build upon this paradigm, and still rely on special tokens and dedicated segmentation decoders. These methods essentially aim to extract semantic embeddings of target objects and then obtain dense segmentation masks by *computing similarity with image features*. Such representations tend to emphasize high-level semantics rather than true pixel-level understanding. HiMTok [57] explores an alternative that removes the reliance on special tokens and SAM-like decoders. However, it still depends on a dedicated mask tokenizer trained on binary masks. Moreover, the expressiveness of the tokenizer is limited and cannot be extended to support other tasks such as image generation. This suggests that segmentation representation in MLLMs remains an open challenge, which we think can be effectively addressed through autoregressive image generation.

**Unified multimodal understanding and generation models** have recently attracted increasing attention for their ability to seamlessly perform both understanding and generation tasks within a single framework. Several works [48, 20, 63, 51] leverage diffusion models for image generation by regressing visual embeddings from MLLM outputs and using them as conditional inputs. TransFusion [77] and Show-O [67] unify next-token prediction and diffusion-based generation within a single transformer framework. Chameleon [49] and Emu3 [59] adopt a shared discrete visual embedding space for both understanding and generation, decoding images through VQ-based tokenizers [19, 71]. Janus [61] decouples the encoder for multimodal understanding and generation, using discrete visual tokens for generation while retaining continuous visual features for better understanding accuracy. VARGPT [78] proposes next-token prediction for understanding and next-scale prediction for image generation, but relies on an additional transformer-based visual decoder.

**Image tokenization** enables discrete outputs from autoregressive models to be reconstructed into images. VQ-VAE [53] encodes images into a downsampled latent space and quantizes the features into discrete token IDs, simplifying the learning process for generative models. VQGAN [19] improves reconstruction quality and training efficiency through adversarial training. TiTok [72] significantly reduces the number of tokens required for image representation, improving generation speed, and further shows that increasing the number of latent tokens consistently enhances reconstruction quality. VAR [50] reformulates visual autoregressive generation as a next-scale prediction task, achieving high efficiency while maintaining a relatively large number of visual tokens.

## 3   Method

In this paper, we propose a novel image segmentation framework based on autoregressive image generation model, using a Vector-Quantized (VQ) autoencoder [53, 19] to tokenize images into discrete tokens and reconstruct them from generated outputs. To address the unique challenges of segmentation, we introduce two key designs. (1) **The MLLM is trained to directly output image tokens**, which is crucial for achieving high pixel-level accuracy. (2) **We utilize a multi-scale generation process that performs coarse-to-fine refinement.** This not only enhances segmentation robustness but also improves inference efficiency. This section first presents the background of the

image tokenizer (Sec. 3.1), then details the architecture (Sec. 3.2), training procedure (Sec. 3.3), and inference process (Sec. 3.4) of our proposed model.

## 3.1 Preliminary

**Vector-Quantized Autoencoder**   The standard VQ model learns to encode images into a latent space and reconstruct them from discrete tokens. Given an input image $\mathbf{I} \in \mathbb{R}^{H \times W \times 3}$, the encoder $\mathcal{E}$ maps it to a latent feature space:

$$f = \mathcal{E}(\mathbf{I}), \quad f \in \mathbb{R}^{\frac{H}{l} \times \frac{W}{l} \times D}, \tag{1}$$

where $l$ is the spatial downsampling factor and $D$ denotes the feature dimesion. The latent features $f$ are then quantized by a vector quantizer $\mathcal{Q}$ into discrete token indices $q \in [V]^{\frac{H}{l} \times \frac{W}{l}}$:

$$q = \mathcal{Q}(f), \quad q^{(i,j)} = \underset{v \in [V]}{\arg\min} \| f^{(i,j)} - c^v \|_2, \tag{2}$$

where $c^v$ is the $v$-th embedding vector in the visual codebook $\mathbb{C} \in \mathbb{R}^{V \times D}$, and $[V]$ denotes the set of codebook indices $\{1, 2, \ldots, V\}$.

The reconstruction of the image can be interpreted as detokenizing discrete visual tokens into an image. In this procedure, the quantized indices $q$ are used to index the corresponding embedding from the visual codebook $\mathbb{C}$, producing the estimated latent feature map $\hat{f}$. The estimated feature map is then passed through the decoder $\mathcal{D}$ to generate the reconstructed image $\hat{\mathbf{I}}$:

$$\hat{f} = \mathbf{lookup}(\mathbb{C}, q), \quad \hat{\mathbf{I}} = \mathcal{D}(\hat{f}). \tag{3}$$

**Multi-Scale VQ Autoencoder**   When using VQ-VAE for autoregressive image generation, the inference process typically requires $\mathcal{O}(n^2)$ steps. To address this inefficiency, VAR[50] introduces a next-scale prediction paradigm for visual token generation. Specifically, the feature map $f$ is quantized into $K$ multi-scale token maps $(r_1, r_2, \ldots, r_K)$, where each map corresponds to a different resolution. At each inference step, the model generates all $h_k \times w_k$ tokens required for the current scale $r_k$ in parallel, repeating this process until $r_K$ reaches the target resolution of $\frac{H}{l} \times \frac{W}{l}$. Moreover, the coarse-to-fine predictions can enhance the generation quality. Based on this paradigm, an image of resolution $256 \times 256$ can be represented using 680 visual tokens, while requiring just $K$ autoregressive inference steps, significantly improving generation efficiency. Given the fast response requirements of image segmentation tasks, we adopts this paradigm to enable efficient autoregressive image generation.

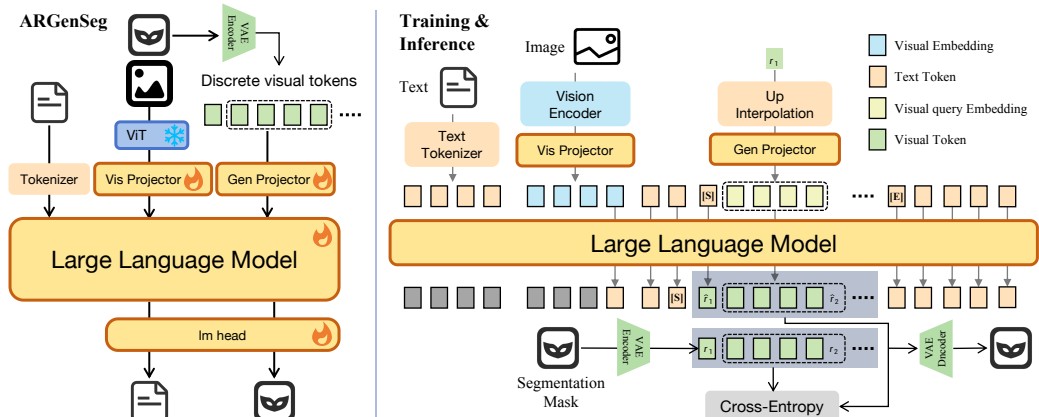

Figure 2: The architecture of ARGenSeg and its training and inference procedures. **Left:** ARGenSeg integrates image segmentation into the MLLM via an autoregressive image generation paradigm. A unified classification prediction head is used to generate both text and visual tokens. **Right:** Visual tokens are generated in parallel using the next-scale prediction strategy. During training, a VAE encoder is used to construct supervision for cross-entropy loss. During inference, the VAE decoder reconstructs the image from the predicted visual tokens. [S]/[E] denotes <gen_start>/<gen_end>.

## 3.2 Architecture

**Multimodal Understanding**   ARGenSeg uses a unified autoregression framework for image understanding and generation as shown in Fig. 2. Our framework employs the built-in tokenizer of the LLM to convert text input into discrete token IDs and corresponding embeddings. For image input, a vision encoder is used to extract features, which are then mapped to the LLM's embedding space via a vision projector. After the concatenated embeddings are fed into the LLM, the model performs next-token prediction to sequentially generate token embeddings. These embeddings are then passed through a classification head to sample discrete token IDs, which are subsequently detokenized into meaningful text. For multimodal understanding tasks, decoupling the framework from image generation preserves the native understanding capabilities of the LLM.

**Image Generation**   To integrate image generation into the framework, we introduce special tokens `<gen_start>` and `<gen_end>` to mark the beginning and end of the generation process. Additionally, the visual token IDs from the visual tokenizer are added to the LLM's vocabulary in the form of `<visual_token_ID>`. When image generation is required, the framework autonomously determines whether to initiate generation based on the input instruction. Upon encountering the `<gen_start>` token, multi-scale image generation begins, where visual tokens for each scale are predicted in parallel. At $k$-th scale, the visual feature corresponding to the visual token map from the previous scale is retrieved by looking up the visual codebook $\mathbb{C}$ and then upsampled to match the resolution of the current scale. A lightweight linear layer, referred to as the generation projector, maps these upsampled visual features into the embedding space of LLM, serving as input for the next scale. This design allows one-step parallel inference to obtain all visual tokens at the current scale. Importantly, the unified prediction head is used to generate visual tokens, which are then directly converted to the corresponding index IDs in the codebook $\mathbb{C}$. Once all visual tokens across scales are generated, they are detokenized by the visual tokenizer to reconstruct the final image.

## 3.3 Training Procedure

**Training Strategy**   In our framework, the vision encoder, large language model, vision projector and classification prediction head are initialized using InternVL 2.5[13], while the multi-scale visual tokenizer is initialized from VAR [50]. During training, the vision encoder and visual tokenizer are kept frozen to reduce the model's reliance on dedicated decoders for pixel-level understanding. By leveraging pre-trained multimodal understanding, the framework converges rapidly when training on image segmentation data. Thus, we employ a single-stage supervised finetuning (SFT) strategy, jointly optimizing both image segmentation and multimodal understanding data. For image generation, we further finetune the pre-trained ARGenSeg model using image generation data to unlock its text-to-image generation capabilities.

**Training Objective**   Since our framework unifies both text and image generation outputs within the LLM codebook, the entire training process is directly supervised using cross-entropy loss, as shown in Fig. 2. During supervision construction, the `<gen_start>` token is added as a marker before image generation begins. The model is expected to **learn both when to initiate image generation and how to generate all the required visual tokens.** The ground-truth visual tokens are obtained using the encoder and quantizer of the VQ-VAE. When constructing input embeddings, the visual tokens for the first scale are obtained by using the `<gen_start>` token as the query. For each subsequent scale, the input embeddings are derived by upsampling the visual token map $r_{k-1}$ of the previous scale to match the size of the current scale. Finally, the `<gen_end>` token is added to ensure the proper progression of subsequent predictions.

## 3.4 Inference

During inference, our model follows a next-token prediction strategy, generating outputs sequentially until the `<gen_start>` token is produced. This token then serves as a query to initiate the generation of visual tokens for the first scale. For the subsequent $K-1$ scales, query embeddings of size $h_k \times w_k$ are obtained by upsampling and projecting the visual token map $\hat{r}_{k-1}$ predicted at the previous scale, enabling parallel generation of all visual tokens at the current scale. Since the upsampling process determines the number of queries, our framework naturally ensures alignment between the number of generated tokens and the input size required by the VQ-VAE decoder. Once the visual tokens for

Table 1: Performance comparison with state-of-the-art methods on three referring image segmentation benchmarks using cIoU. (ft) indicates models further finetuned on RefCOCO/+/g after mixed training.

| Paradigm | Method | RefCOCO | | | RefCOCO+ | | | RefCOCOg | |
|---|---|---|---|---|---|---|---|---|---|
| | | val | testA | testB | val | testA | testB | val | test |
| Boundary Point-based | PolyFormer-B [42] | 74.8 | 76.6 | 71.1 | 67.6 | 72.9 | 59.3 | 67.8 | 69.1 |
| | VistaLLM-7B [42] | 74.5 | 76.0 | 72.7 | 69.1 | 73.7 | 64.0 | 69.0 | 70.9 |
| Dedicated Segmentation Head-based | LISA-7B(ft) [29] | 74.9 | 79.1 | 72.3 | 65.1 | 70.8 | 58.1 | 67.9 | 70.6 |
| | PixelLM-7B [45] | 73.0 | 76.5 | 68.2 | 66.3 | 71.7 | 58.3 | 69.3 | 70.5 |
| | GSVA-7B [65] | 76.4 | 77.4 | 72.8 | 64.5 | 67.7 | 58.6 | 71.1 | 72.0 |
| | GSVA-7B(ft) | 77.2 | 78.9 | 73.5 | 65.9 | 69.6 | 59.8 | 72.7 | 73.3 |
| | LaSagnA-7B [60] | 76.8 | 78.7 | 73.8 | 66.4 | 70.6 | 60.1 | 70.6 | 71.9 |
| | VisionLLM v2 [62] | 76.6 | 79.3 | 74.3 | 64.5 | 69.8 | 61.5 | 70.7 | 71.2 |
| | OMG-LLAVA [73] | 75.6 | 77.7 | 71.2 | 65.6 | 69.7 | 58.9 | 70.7 | 70.2 |
| | OMG-LLAVA(ft) | 78.0 | 80.3 | 74.1 | 69.1 | 73.1 | 63.0 | 72.9 | 72.9 |
| | GLaMM [44] | 79.5 | 83.2 | 76.9 | 72.6 | 78.7 | 64.6 | 74.2 | 74.9 |
| | u-LLAVA [68] | 83.0 | 85.1 | 80.5 | 77.1 | 81.7 | 70.6 | 77.1 | 78.0 |
| | PSALM [75] | 83.6 | 84.7 | 81.6 | 72.9 | 75.5 | 70.1 | 73.8 | 74.4 |
| | GroundHog-7B [74] | 78.5 | 79.9 | 75.7 | 70.5 | 75.0 | 64.9 | 74.1 | 74.6 |
| | SAM4MLLM-8B [42] | 79.8 | 82.7 | 74.7 | 74.6 | 80.0 | 67.2 | 75.5 | 76.4 |
| | LMM$_{\text{HiMTok}}$-8B [57] | 81.1 | 81.2 | 79.2 | 77.1 | 78.8 | 71.5 | 75.8 | 76.7 |
| | LMM$_{\text{HiMTok}}$-8B(ft) | 85.0 | 85.2 | **83.5** | 79.7 | 82.7 | 76.0 | 80.0 | 80.6 |
| Generation based | ARGenSeg | 82.2 | 84.0 | 80.1 | 77.9 | 81.8 | 73.3 | 78.4 | 79.6 |
| | ARGenSeg (ft) | **86.3** | **87.5** | 82.7 | **82.3** | **85.8** | **77.0** | **81.7** | **83.5** |

all $K$ scales are obtained, the VAR tokenizer decodes them into the final image. To ensure smooth progression of subsequent inference, the <gen_end> token is manually added.

# 4 Experiments

## 4.1 Experimental Setup

**Datasets**  As described in Sec. 3.3, we perform a single-stage supervised finetuning to jointly train on both image segmentation and multimodal understanding data. Details of all datasets used are provided in Appendix A. The training of ARGenSeg relies entirely on publicly available external datasets. Specifically, we use 402K image segmentation samples, which are significantly fewer than the 2.91M samples used by HiMTok[57] and constitute a strict subset of their data. For multimodal understanding, we use 1.25M samples derived from the open-source dataset of InternVL 1.2 [14].

**Implementation Details**  Our model accepts input images of arbitrary resolutions, while the output images are generated at the resolution of $256 \times 256$. The image tokenizer uses a downsampling ratio $l = 16$, with a feature dimension $D = 32$ and a visual codebook size $V = 4096$. The model operates with $K = 10$ scales. During training, we use the AdamW [36] optimizer with a maximum learning rate of $4 \times 10^{-5}$ and employ cosine learning rate scheduling. The batch size is set to 128.

## 4.2 Referring Segmentation

**Referring Expression Segmentation**  Recent works have increasingly focused on equipping multimodal large language models with image segmentation capabilities, aiming to leverage their strong language understanding for more complex segmentation tasks. Referring Expression Segmentation (RES) requires models to segment target objects in an image based on natural language descriptions. We evaluate our approach on standard RES benchmarks RefCOCO/+/g [37, 70]. Following prior works [29, 57], we assess two versions of our model: one trained on the mixed dataset, and another further finetuned on the in-domain training sets of RefCOCO/+/g. As shown in Tab. 1, our method consistently outperforms the previous state-of-the-art, HiMTok [57], across both versions, despite training on fewer segmentation data. It is worth noting that, our approach achieves superior results without relying on a dedicated segmentation head, demonstrating the effectiveness of our unified multimodal understanding and generation framework.

Table 2: Performance comparison with state-of-the-art methods on generalized referring expression segmentation. * indicates zero-shot performance.

| Method | val | | testA | | testB | | Average |
|---|---|---|---|---|---|---|---|
| | cIoU | gIoU | cIoU | gIoU | cIoU | gIoU | |
| LISA-7B [29] | 38.7 | 32.2 | 52.6 | 48.5 | 44.8 | 39.7 | 42.8 |
| LISA-7B(ft) | 61.8 | 61.6 | 68.5 | 66.3 | 60.6 | 58.8 | 62.9 |
| GSVA-7B [65] | 61.7 | 63.3 | 69.2 | 70.1 | 60.3 | 61.3 | 64.3 |
| GSVA-7B(ft) | 63.3 | 66.5 | 69.9 | 71.1 | 60.5 | 62.2 | 65.6 |
| LaSagnA* [60] | 38.1 | 32.4 | 50.4 | 47.3 | 42.1 | 38.9 | 41.5 |
| PSALM* [75] | 42.0 | 43.3 | 52.4 | 54.5 | 50.6 | 52.5 | 49.2 |
| GroundHog-7B [74] | - | 66.7 | - | - | - | - | 66.7 |
| SAM4MLLM-8B [42] | 67.8 | 71.9 | 72.2 | **74.2** | 63.4 | 65.3 | 69.1 |
| LMM$_{HiMTok}$-8B [57] | 66.8 | 68.7 | 68.6 | 67.6 | 65.8 | 64.1 | 66.9 |
| ARGenSeg | **72.2** | **74.7** | **73.6** | 73.7 | **70.0** | **70.4** | **72.4** |

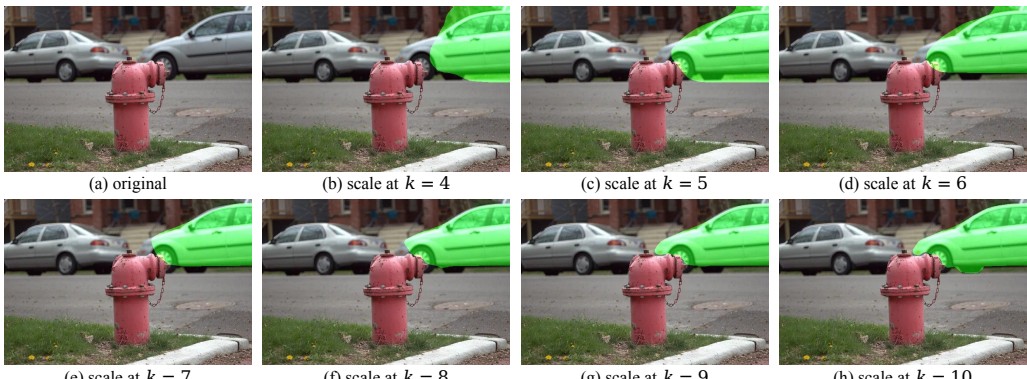

(a) original  (b) scale at $k = 4$  (c) scale at $k = 5$  (d) scale at $k = 6$

(e) scale at $k = 7$  (f) scale at $k = 8$  (g) scale at $k = 9$  (h) scale at $k = 10$

Figure 3: Multi-scale generation process of the segmentation mask. The model first localizes the target object and then progressively refines its boundaries.

Fig. 3 illustrates the multi-scale mask generation process of ARGenSeg. The model first locates the target object and then progressively refines the segmentation boundaries. This coarse-to-fine reasoning process aligns with human intuition and enhances the robustness of image segmentation.

**Generalized Referring Expression Segmentation**  We further evaluate our model on the more challenging gRefCOCO benchmark [32], where segmentation instructions may refer to multiple objects or none at all. As shown in Tab. 2, our method outperforms all prior approaches that rely on dedicated segmentation heads, highlighting the strong understanding and segmentation capabilities of our unified framework.

## 4.3  Multumodal Understanding

Our model adopts InternVL 2.5 [13] as the underlying MLLM and is finetuned on both understanding and segmentation data. To fairly assess the effect of adding segmentation supervision on the model's understanding capability, we finetune a baseline using only understanding data. We evaluate the model's understanding performance using two tasks. The first task is visual grounding, where we use the RefCOCO/+/g datasets for referring expression comprehension (REC). As shown in Tab. 3, our model successfully retains and even slightly enhances its grounding ability while acquiring segmentation capabilities. The second task evaluates object hallucination in MLLMs using POPE [30] as the benchmark. Results in Tab. 3 also demonstrate a performance improvement of our model compared to the baseline. These results highlight the effectiveness of our proposed framework in unifying understanding and segmentation tasks. A further discussion on the understanding performance is provided in Appendix C.1.

Table 3: Multimodal understanding performance compared with the baseline. * indicates further finetuning on understanding data.

| Method | RefCOCO | | | RefCOCO+ | | | RefCOCOg | | POPE |
|---|---|---|---|---|---|---|---|---|---|
| | val | testA | testB | val | testA | testB | val | test | |
| InternVL2.5-8B* [13] | 89.0 | 92.6 | 84.3 | 83.4 | **89.1** | **76.5** | 83.5 | 85.0 | 86.73 |
| ARGenSeg | **89.6** | **92.8** | **84.4** | **83.8** | 88.8 | **76.5** | **86.1** | **85.6** | **87.57** |

## 4.4 Function Extension

**Interactive Segmentation**    Interactive segmentation allows users to provide diverse input prompts during segmentation tasks to meet varying application needs. We finetune ARGenSeg on the COCO-Interactive dataset [75] to unlock its interactive segmentation capabilities. During training, various forms of interactive prompts are used, including **points, scribbles, and bounding boxes.** Bounding boxes are provided as textual input to the MLLM, while points and scribbles are represented as binary masks and fed in as additional visual inputs. We observe that, building upon pre-trained segmentation capabilities, the model quickly adapts to interactive segmentation tasks. Qualitative results are shown in the top portion of Fig. 4, while the quantitative evaluation can be found in the Appendix C.2.

**Image Generation**    Our model leverages a universal image tokenizer, enabling the potential for image generation. We finetune ARGenSeg on 1.28M class-based samples from the ImageNet-Instruct-class dataset [78], using a batch size of 512 for $20k$ iterations. This successfully enables class-conditional image generation, as illustrated in Fig. 1. We then continue training for an additional $30k$ iterations with a batch size of 256 on the ImageNet-Instruct1270K dataset [78], which is based on instruction-conditioned generation. The results of instruction-based image generation are shown in the bottom of Fig. 4. Notably, our model achieves these results **without relying on pre-trained generation model**, using only a small amount of data and training iterations.

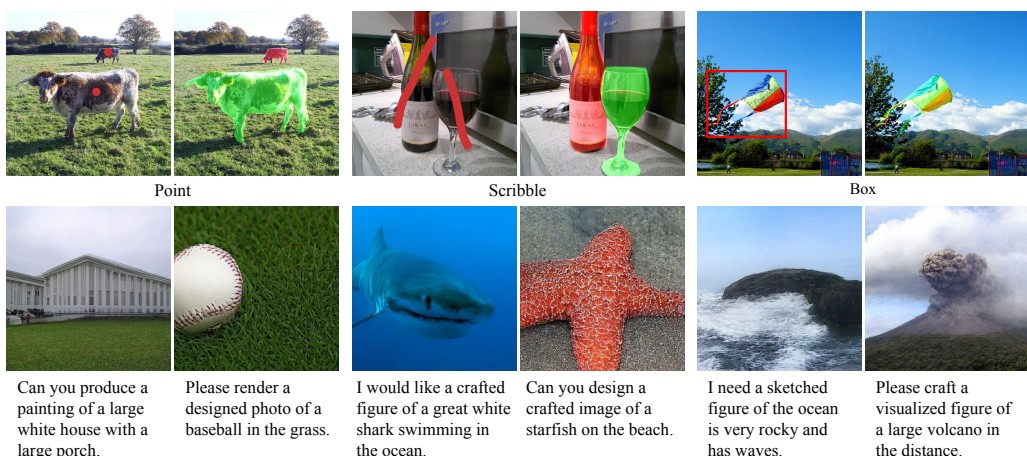

Figure 4: **Top:** Visualization of interactive segmentation. Points and scribbles are provided as visual prompts, while bounding boxes are input via text. **Bottom:** Visualization results of instruction-based image generation. The model is trained on image generation data for only $50k$ iterations.

## 4.5 Efficiency Analysis

We compare ARGenSeg with previous autoregressive generation models and MLLM-based segmentation methods in terms of inference time required to generate a $256 \times 256$ image or mask. All experiments are conducted using official implementations on an NVIDIA A100 GPU. Segmentation performance is evaluated using cIoU on RefCOCO-val. Detailed results are provided in Tab. 4.

Compared to sequential token generation approaches such as Emu3 [59], our parallel inference achieves more than $10\times$ speedup. While VARGPT [78] also employs VAR as its visual tokenizer,

Table 4: Computational efficiency comparison. "Num." represents the number of required tokens. Time is tested by seconds per image.

| Method | Paradigm | Num. | cIoU | Time |
|---|---|---|---|---|
| Emu3 [59] | VQ-GAN[19] | 1024 | - | 59.4 |
| VARGPT [78] | VAR + Vis.Dec | 680 | - | 2.64 |
| PixelLM [45] | Query + Seg.Dec | 6 | 73.0 | 0.91 |
| HiMTok [57] | Mask Tokenizer | 32 | 81.1 | 1.89 |
| ARGenSeg | VAR Tokenizer | 680 | 82.2 | 1.28 |

Table 5: Ablation study on the impact of understanding capability, pretraining-stage, and generation projector on segmentation performance.

| Experiment | Ref. | Ref.+ | Ref.g | Average |
|---|---|---|---|---|
| Baseline | 82.2 | 77.9 | 78.4 | 79.5 |
| Only-Seg. | 80.5 | 73.8 | 73.2 | 75.8 |
| Gen Projectetor | 80.5 | 73.7 | 73.4 | 75.9 |
| Pretrain | 80.8 | 74.9 | 74.1 | 76.6 |

Table 6: Ablation study on the visual tokenizer. All results are reported on the val splits, using gIoU (per-sample IoU averaged over the dataset) as the segmentation metric.

| Tokenizer Type | Prediction | RefCOCO | RefCOCO+ | RefCOCOg | Average | Time |
|---|---|---|---|---|---|---|
| Single-scale | next-token | 82.1 | 71.8 | 65.8 | 73.23 | 5.50 s |
| Multi-scale | next-scale | 80.5 | 76.7 | 70.4 | 75.87 | 1.28 s |

our method is approximately $2\times$ more efficient, due to its simplified architecture. In contrast to VARGPT, our model directly uses the classification head to predict token IDs from the VAR codebook, eliminating the need for an additional transformer-based visual decoder. PixelLM [45], a identifier-based approach, uses only six tokens and a dedicated segmentation decoder, making it slightly faster than ARGenSeg. However, its segmentation performance is significantly lower. While HiMTok [57] employs a dedicated mask tokenizer to achieve notable segmentation performance using only 32 visual tokens for efficiency, our method achieves superior performance while offering a clear advantage in inference speed.

## 4.6 Ablation Study

**Ablation on Understanding Data**    We compare our baseline, fine-tuned on both understanding and segmentation data, against a counterpart trained solely on segmentation data. As shown in Tab. 5, incorporating understanding data significantly improves performance on reasoning-based segmentation, particularly on the semantically challenging RefCOCO+/g dataset. This highlights the value of unifying segmentation with a multimodal large language model.

**Ablation on Model Architecture and Training Strategy**    We analyze the effects of model architecture and training strategy. First, to ablate the architecture, we replace our default single-layer generation projector with a two-layer variant. Results indicate that the simpler design is sufficient. Second, to assess the training strategy, we introduce a pre-training phase where only the generation projector is trained, followed by a full fine-tuning stage. As shown in Tab. 5, this two-stage approach offers only marginal gains on RefCOCO+/g and little impact on RefCOCO, while increasing training complexity. Therefore, for efficiency, our final model adopts a direct, single-stage fine-tuning strategy.

**Ablation on Visual Tokenizer**    We ablate our multi-scale visual tokenizer by comparing it against a single-scale tokenizer, for which we adopt the pre-trained VQ-GAN [59] from Janus [61]. As shown in Tab. 6, using multi-scale scheme not only demonstrates a clear speed advantage but also improves robustness through its inherent coarse-to-fine refinement process. Further ablations, including an analysis of using semantic embeddings instead of visual tokens, are provided in Appendix D.

## 5    Conclusion

In this paper, we present ARGenSeg, a unified framework that integrates image segmentation into multimodal large language models through an image generation paradigm. To address the unique challenges of segmentation, we design the framework so that the MLLM directly outputs image tokens for pixel-level accuracy and utilizes multi-scale image generation for high responsiveness and robustness through coarse-to-fine refinement. Our experiment results are the first to show that unified MLLM models can perform state-of-the-art segmentation without any extra task-specific segmentation heads, providing an effective technical pathway for unified AGI.

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

# Appendix of ARGenSeg

## A Implementation Details

**Datasets**   The datasets used for image segmentation, multimodal understanding, and image generation are listed in Tab. 7. To ensure a fair comparison, we exclusively use subsets of the data employed by the previous state-of-the-art method, HiMTok [57]. Specifically, we train on 402K segmentation samples compared to HiMTok's 2.91M, and 1.25M multimodal understanding samples compared to HiMTok's 4.2M. Image generation data are used only in the optional function-extension stage.

Table 7: Training data used in our experiments.

| Task | Datasets |
|---|---|
| Image Segmentation | ADE20K(20K) [76], COCO-Panoptic(118K) [31], gRefCOCO (79K) [32], RefCOCO/+/g(127K) [37, 70], LISA++ Inst.Seg(58K) [69] |
| Multimodal Understanding | AI2D [25], ChartQA[38], COCO-Text[55], DocVQA[18], LLaVA-150K[34], GQA[23], DVQA[24], OCR-VQA[39], TextVQA[47], SynthDoG-EN [26], InternVL-SA1B-Caption [14], VisualGenome [28], GeoQA+[10] |
| Image Generation | ImageNet-Instruct-class [78], ImageNet-Instruct1270K [78] |

**Inference Details**   During inference, we get visual outputs exclusively from the logits corresponding to visual tokens in the MLLM codebook. This constraint ensures compatibility with the visual tokenizer and enables successful reconstruction of the image. For image segmentation tasks, we adopt a deterministic **argmax** sampling strategy to obtain the predicted visual tokens. For image generation tasks, we apply classifier-free guidance (CFG) to compute the output distribution over visual tokens, followed by **top-k** sampling to enhance the diversity and quality of generated images.

## B Additional Qualitative Results

**Multi-scale Image Generation**   We provide visualization of segmenting similar objects in the same image using different instructions, as shown in Fig.5. From the multi-scale mask generation process, it is evident that our model can correctly understand and localize the target based on the given instructions. The ability to correctly follow distinct segmentation commands indicates that ARGenSeg possesses a robust understanding of both spatial positions and semantic relationships.

**Comparison with Single-scale Generation**   We compare our method with HiMTok [57], treating it as a representative single-scale generative segmentation approach. We conducted a thorough evaluation on the test set and visualized cases where ARGenSeg succeeds while HiMTok fails. As shown in Fig. 6, these cases reveal two primary advantages of our coarse-to-fine, multi-scale generation scheme: **(1) Robust Target Identification in Multi-object Scenarios.** The initial coarse localization stage effectively identifies the target object even when multiple similar objects are present. **(2) Enhanced Mask Quality through Progressive Refinement.** Following target identification, the multi-scale refinement process progressively improves mask precision for higher-quality segmentation. For instance, in the case of a partially occluded teddy bear, both HiMTok and our coarse localization stage initially segment only a visible part. However, our model's subsequent fine-grained refinement successfully reconstructs the entire object while correctly excluding the occluder.

## C Additional Quantitative Results

### C.1 Performance on Multimodal Understanding

We further assess the multimodal understanding capabilities of ARGenSeg. As shown in Tab. 8, the inclusion of segmentation data does not cause the model to lose its reasoning capability. While we

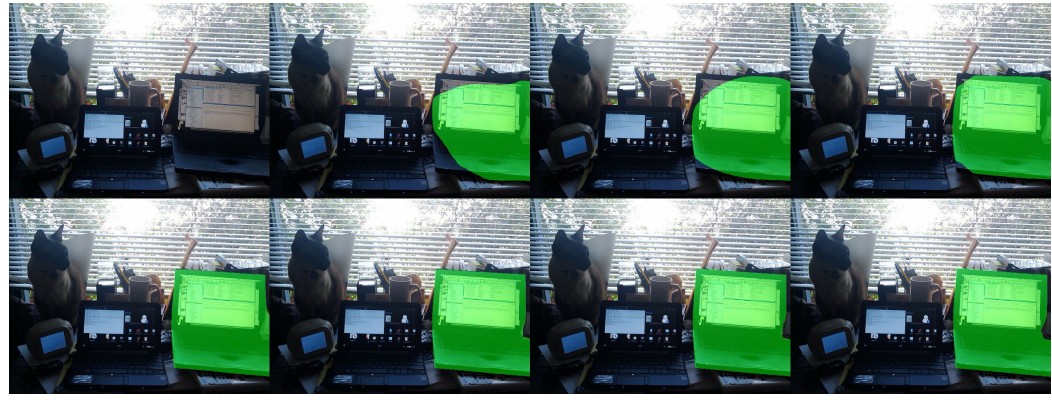

computer on right side of picture

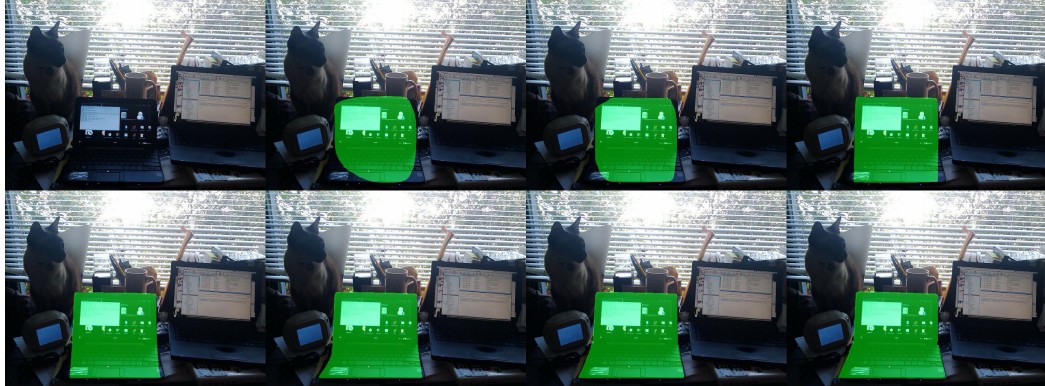

laptop next to cat

Figure 5: Visualization of using different segmentation instructions in the same image.

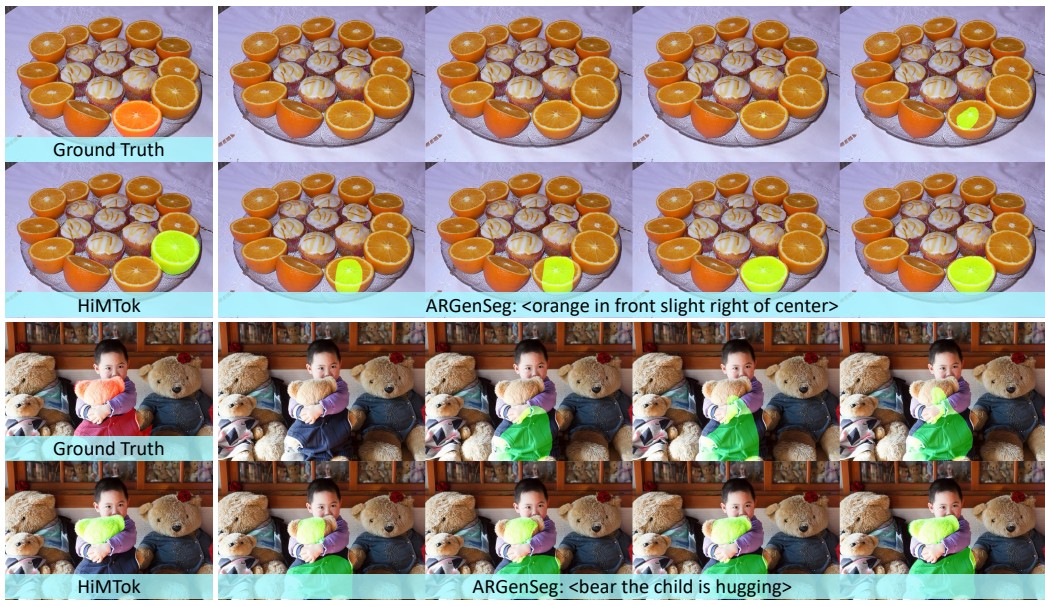

Figure 6: Comparison between multi-scale and single-scale generative segmentation approach. The examples highlight scenarios where the multi-scale approach excels.

observe slight performance drops on some benchmarks, we attribute this minor degradation not to the segmentation task itself, but to the significantly smaller and lower-quality understanding corpus used for fine-tuning (1.25M vs. the 16.3M samples used for InternVL-2.5 [13]). To validate this hypothesis, we conducted a control experiment: fine-tuning InternVL-2.5 solely on the same understanding data for an increasing number of steps. The performance declined monotonically, mirroring the trend observed with joint segmentation training and thus confirming our attribution.

Table 8: Multimodal understanding results across benchmarks.

| Method | POPE | TextVQA | VQAv2 | MMMU-val | AI2D |
|---|---|---|---|---|---|
| InternVL2.5-ft-1ep | 86.73 | 63.54 | 80.40 | 43.7 | 78.7 |
| InternVL2.5-ft-4ep | 86.01 | 59.73 | 79.28 | 36.8 | 74.8 |
| ARGenSeg | 87.57 | 56.98 | 77.87 | 33.4 | 69.6 |

## C.2 Results on Interactive Segmentation

To ensure a fair comparison with HiMTok, which was not trained on interactive-segmentation data, we omitted this task from our main experiments. Here, we evaluate our model on the COCO-Interactive benchmark [75], reporting the cIoU metric. It is worth noting that while PSALM [75] was fine-tuned for 10 epochs according to its official implementation, our model is fine-tuned for only a single epoch due to computational constraints. As shown in Tab. 9, ARGenSeg significantly outperforms SAM [27] in interactive segmentation. Moreover, it achieves performance comparable to PSALM with substantially less fine-tuning, which underscores the strong generalization capabilities of our model.

Table 9: Quantitative results on interactive segmentation. The results for SAM and PSALM are sourced directly from the PSALM paper.

| Method | Point | Scribble | Box |
|---|---|---|---|
| SAM-B [27] | 33.6 | – | 68.7 |
| SAM-L [27] | 37.7 | – | 71.6 |
| PSALM [75] | 74.0 | 80.0 | 80.9 |
| ARGenSeg | 65.6 | 68.6 | 79.1 |

# D  Additional Ablation Studies

Table 10: Ablation study of MLLM backbones and image generation strategies. The segmentation performance is measured in cIoU.

| Method | Backbone | Generation Strategy | RefCOCO | RefCOCO+ | RefCOCOg |
|---|---|---|---|---|---|
| HiMTok [57] | InternVL-2.5 | Single-scale VQ | 81.1 | 77.1 | 75.8 |
| ARGenSeg-LLaVA | LLaVA–1.5 | Multi-scale VQ | 72.7 | 68.3 | 69.1 |
| ARGenSeg-InternVL | InternVL-2.5 | Multi-scale VQ | **82.2** | **77.9** | **78.4** |
| ARGenSeg-DiT | InternVL-2.5 | Diffusion Head | 59.0 | 62.7 | 64.1 |

## D.1  Ablation on MLLM Backbone

Our approach, which integrates a VQVAE codebook into the MLLM's token space, is designed to be model-agnostic. To demonstrate this portability, we replaced the default InternVL-2.5 backbone with LLaVA-1.5 [33], a LLaMA-2-based MLLM. As shown in Tab. 10, our pipeline successfully imparts segmentation capabilities to LLaVA-1.5.

As established in Sec. 4.6, referring segmentation performance is highly correlated with the MLLM's underlying understanding ability. Consequently, given LLaVA-1.5's weaker understanding capabilities compared to InternVL-2.5, the resulting segmentation performance is expectedly lower. Nevertheless, Tab. 10 shows that with the same powerful InternVL-2.5 backbone, our method outperforms HiMTok. This confirms that our performance gains are inherent to our approach and not merely a byproduct of a stronger backbone.

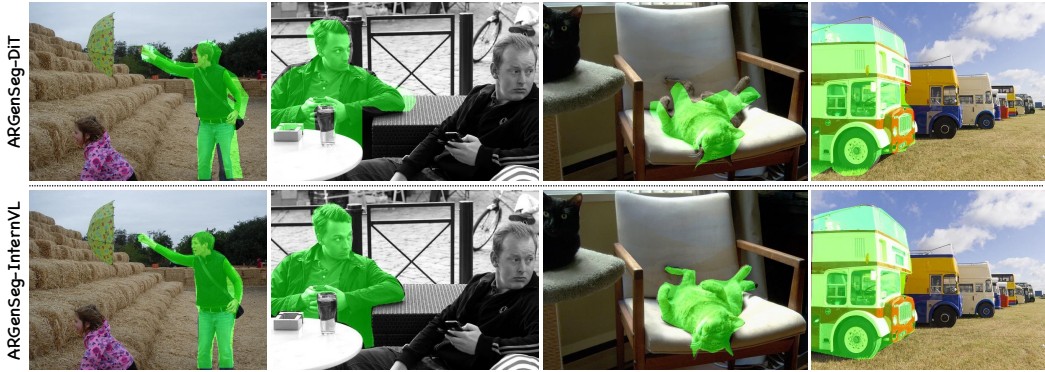

Figure 7: Comparison between direct visual token generation and DiT-based generation. The DiT-based approach, which uses semantic embeddings from the MLLM, struggles with pixel-level accuracy, leading to artifacts like spatial shifts and imprecise boundaries.

## D.2 Ablation on Image Generation Strategy

To further validate our choice of generation strategy, we explore an alternative approach where the MLLM outputs semantic embeddings to a separate diffusion head (DiT) for segmentation, inspired by MetaQuery [41]. Specifically, we configure the MLLM to generate learnable queries, which are then mapped to the feature space of the pre-trained SANA-1.5 1.6B [66] via a connector module.

This alternative strategy, labeled as ARGenSeg-DiT in Tab. 10, led to a severe performance degradation. As show in Fig. 7, while the model could roughly localize the target region, the generated masks suffered from significant artifacts, such as spatial shifts and inflation, indicating poor pixel-level accuracy. This experiment underscores the importance of the MLLM directly generating discrete image tokens to maintain the high pixel-level precision crucial for segmentation tasks.

## E   Limitations

This paper proposes a novel image segmentation paradigm based on autoregressive image generation, integrating multimodal understanding, generation, and image segmentation into a unified framework. Our model demonstrates strong performance across a range of segmentation tasks, and further shows the potential to extend to more complex scenarios, such as interactive segmentation and text-to-image generation. The unified framework also shows promise for expanding to broader tasks, such as image editing and depth estimation. However, due to resource constraints, exploring these extensions is beyond the scope of this work, and we consider them as promising directions for future research.

## F   Broader Impacts

This work contributes to the development of unified multimodal frameworks by integrating dense image segmentation into the unified multimodal understanding and generation models. The proposed framework may inspire future research toward more generalizable, modular, and efficient visual-language models that require fewer task-specific components. Potential applications include human-robot interaction, assistive vision systems, and real-world visual understanding under low supervision. However, like most large-scale models, ARGenSeg may inherit biases from pre-trained components or datasets. Care should be taken to evaluate fairness and robustness when deploying it in real-world scenarios, especially in sensitive domains such as healthcare or surveillance.

