# OpenReview forum: "ARGenSeg: Image Segmentation with Autoregressive Image Generation Model"
_NeurIPS.cc/2025/Conference — NeurIPS 2025 poster_

### Official Review · Reviewer_TNWY · 2025-07-01

**Clarity:** 3
**Significance:** 3
**Originality:** 2
**Rating:** 4
**Confidence:** 3

**Summary:**

This paper presents ARGenSeg, a unified framework for image segmentation and generation based on a multimodal large language model. The approach discretizes images and segmentation masks into visual tokens using VQ-VAE, and generates these tokens via VAR. The model produces token sequences for both segmentation and image generation. Experimental results demonstrate reasonable improvements over existing methods.

**Questions:**

Please refer to the weaknesses for the rebuttal.

**Ethical Concerns:**

["NO or VERY MINOR ethics concerns only"]

**Final Justification:**

The authors' rebuttal addressed some of my concerns by providing additional experiments and clarifications. Therefore, I am keeping my current score of borderline accept.

**Limitations:**

Yes

**Quality:**

3

**Strengths And Weaknesses:**

Strengths

1. The paper is well written, and the figures and tables are informative.

2. The proposed framework provides some inspiration for segmentation tasks by unifying segmentation and generation within a multimodal model.

3. The method achieves reasonable improvements in segmentation performance and inference speed compared to previous methods.

Weaknesses

1. The paper lacks deep insights. The concept of a unified framework for visual generation and understanding is not novel, and the use of VAR for acceleration is adapted from existing work. The segmentation aspect is not thoroughly explored, as the method appears to treat the segmentation mask as a special case of image generation, but this is not discussed in detail. As a result, the paper lacks an in-depth analysis specific to the unique challenges of segmentation tasks.

2. The Function Extension section lacks thorough quantitative analysis. For example, for interactive segmentation, it would be helpful to compare the proposed method with SAM to better assess ARGenSeg’s generalizability.

3. There is a lack of experimental analysis on high-resolution images. The experiments primarily focus on low-resolution images and do not address the performance of the method on high-quality segmentation over high-resolution images.

---

> ### Author Rebuttal · Authors · 2025-07-30
>
> Thank you for your positive assessment of our method. We would like to address your concerns as follows.
>
> ### **W1: deep insights and architecture novelty**
>
> **A1:**
> We sincerely apologize for any confusion regarding the depth of our architectural insights. In the final version we will expand the discussion to make these points explicit.
>
> We did not simply treat segmentation as yet another generation task and fine-tune an existing unified model. Instead, **we redesigned the unified MLLM around the distinctive challenges of dense perception.**
> In the course of our research, we have carefully considered the unique challenges of image segmentation, which informed several key design choices in our model:
>
> **1. The strong understanding capabilities of MLLMs are particularly beneficial for image segmentation.**
> - Therefore, we **utilize continuous visual features as input** to preserve the MLLM's strong understanding ability (see L36-37 of the paper).
>
> **2. image segmentation requires higher accuracy and precision at the low-level, pixel-wise scale, whereas standard image generation focuses more on semantic alignment.**
> - Rather than adding an external VAR-style transformer as a generation head (as in VARGPT), or letting the MLLM produce only semantic conditions for downstream models, we **force the MLLM itself to generates the image tokens.**(L42-44, L83-84).
>    - We conducted experiments where the MLLM only outputs semantic conditions to DiT for image generation. The results show that, while the generated segmentation roughly locates target regions, the outputs often suffer from noticeable inflation or shifts—indicative of significant pixel-level accuracy and consistency issues.
>    - The technical report of VARGPT-v1.1[1] discusses image editing but only experiments with style transfer. The visual results (Fig. 9) display marked consistency issues, which we believe stem from the same limitation.
>
> **3. As a perceptual task, image segmentation has higher requirements for response speed.**
> - Therefore, we **adopt a multi-scale image-generation scheme that predict all visual tokens of the current scale in parallel.** (discussed in the Introduction and Table 6: L45-50, L83-84).
>    - A single-scale VQGAN baseline slows inference **from 1.28 s to 5.50 s**, confirming the necessity of this design.
>
> **4. Our goal is to integrate not only multimodal understanding and generation but also visual perception tasks into a unified AGI framework.**
> - For this reason, we do not use specialized segmentation heads such as SAM. Instead, we address image segmentation through an autoregressive image generation approach.
> - Our research demonstrates the feasibility of incorporating dense perceptual tasks into a unified understanding and generation framework, and achieves higher performance compared to approaches requiring additional specialized segmentation heads.
>   - This provides an effective technical pathway and practical foundation for the development of unified AGI models (as discussed in the Introduction: L24-25).
>
> To the best of our knowledge, **we are the first to integrate multi-scale VQ-VAE visual indices directly into an MLLM’s output space for a unified understanding-and-generation framework.** This design choice was conceived independently and precedes the public release of VARGPT; it was driven by the challenges above rather than adopted from prior work.
> - **We hope this clarifies the novelty of our approach and inspires further research on unified models.**
>
> ### **W2: Quantitative result for Interactive segmentation**
>
> **A2:**
> Thank you for this insightful suggestion. We will add the requested quantitative results to the final version.
>
> **1.  The quantitative results for interactive segmentation show that our method significantly outperforms SAM.**
>
> Because the training set of our main baseline (HiMTok [2]) does not contain interactive-segmentation data, we excluded such data from the main experiments to ensure a fair comparison; we only provided a qualitative proof-of-concept in the “Capability Extension” section. Following your advice, we now evaluate on the COCO-Interactive benchmark and report cIoU below. Results for SAM and PSALM are taken from Table 10 of the PSALM [3] paper. Please note that, according to the official code, PSALM was fine-tuned for 10 epochs, whereas we continued fine-tuning for only 1 epoch due to limited compute.
> - The experimental results in Table 1 show that our model significantly outperforms SAM in interactive segmentation. Furthermore, our method achieves performance comparable to PSALM with much less training, which demonstrates the strong generalization ability of ARGenSeg.
>
> Table1 (Quantitative evaluation of interactive segmentation)
> | Method   | Point | Scribble | Box   |
> |----------|-------|----------|-------|
> | SAM-B    | 33.6  | –        | 68.7  |
> | SAM-L    | 37.7  | –        | 71.6  |
> | PSALM    | 74.0  | 80.0     | 80.9  |
> | ARGenSeg | 65.6  | 68.6     | 79.1  |
>
> Regarding image generation, our model learns the generative component from scratch (in contrast to VARGPT, which uses a pre-trained generator). Owing to computational constraints, we stopped training once we verified that the model could indeed unlock image-generation capability.
> We leave a more thorough exploration of the image generation ability of our model for future work.
>
> ### **W3: experimental analysis on high-resolution images.**
>
> **A3:**
> Thank you for this valuable suggestion. In the final version we will add discussion on high-resolution segmentation.
>
> **1. ARGenSeg natively supports variable-resolution inputs up to 4 K**
>
> In our experiments, the model accepts image inputs of arbitrary resolution, with the only restriction being that the output segmentation mask is set to a resolution of $256\times256$.
>
> - Our MLLM backbone natively supports variable-resolution inputs up to 4 K, so ARGenSeg can directly process high-resolution images without extra preprocessing.
> - Following your advice, we evaluated ARGenSeg on images at $ 2344 \times 3190 $ resolution. Qualitative results show that the model produces accurate segmentation masks on these high-resolution inputs, confirming its scalability.
>
> [1] Zhuang X, Xie Y, Deng Y, et al. Vargpt-v1. 1: Improve visual autoregressive large unified model via iterative instruction tuning and reinforcement learning[J]. arXiv preprint arXiv:2504.02949, 2025.
>
> [2] Wang T, Cheng C, Wang L, et al. Himtok: Learning hierarchical mask tokens for image segmentation with large multimodal model[J]. arXiv preprint arXiv:2503.13026, 2025.
>
> [3] Zhang Z, Ma Y, Zhang E, et al. Psalm: Pixelwise segmentation with large multi-modal model[C]//European Conference on Computer Vision. Cham: Springer Nature Switzerland, 2024: 74-91.

---

> > ### Comment · Reviewer_TNWY · 2025-08-05
> >
> > Thank you for the detailed rebuttal, which has addressed some of my concerns.
> >
> > However, I maintain my core point that the paper should elaborate on the segmentation-specific designs in the technical sections. Your response indicates that much of the discussion on segmentation is limited to the introduction, and the technical part still needs to be strengthened. It would be necessary to support your arguments with concrete examples.
> >
> > Overall, the paper has technical contributions, and I will keep my current score.

---

> > > ### Author Response · Authors · 2025-08-07
> > >
> > > Dear Reviewer TNWY,
> > >
> > > Thank you for your recognition and your recommendation for acceptance. We sincerely appreciate your thoughtful review and are pleased to have resolved some of your concerns.
> > >
> > > We will adopt your suggestions to further strengthen the technical discussion in the final version, and will support our findings with more concrete examples. Specifically:
> > >
> > > **1. By enabling the MLLM to directly output semantic conditions to DiT for image segmentation, we demonstrate the importance of direct image generation by the MLLM for low-level accuracy, which is crucial for segmentation tasks.**
> > > - We refer to MetaQuery [1] and use learnable queries to let the MLLM output semantic embeddings, which are then mapped to the DiT feature space via a connector module, and then use DiT to perform generative image segmentation.
> > >   - For DiT, we use SANA-1.5 1.6B [2].
> > > - After training, we observed that:
> > >    - The quantitative segmentation scores dropped to around 50, showing a severe performance drop.
> > >    - Qualitative visualization of the segmentation masks shows that, while the model could roughly localize the target region, there were evident shifts or mask inflation issues, highlighting significant problems in pixel-level accuracy and consistency.
> > > - We will include these results and visualizations in the final submission to clarify our findings.
> > >
> > >
> > > **2. We added an ablation experiment replacing the proposed multi-scale image generation with a single-scale next-token prediction, to demonstrate the speed advantage of multi-scale generation, and also found that the coarse-to-fine paradigm of multi-scale brings stronger segmentation robustness.**
> > > - We replaced our multi-scale VQ tokenizer with the single-scale VQ-GAN pretrained in Janus [3], and trained it on the same pure segmentation data. We report the segmentation metric "giou," which represents the per-sample IoU averaged over the dataset.
> > > - As shown in Table 1, the results demonstrate that:
> > >    - The multi-scale generation paradigm has a significant speed advantage (1.28s Vs 5.50s).
> > >    - As visualized in Fig. 3, due to its coarse-to-fine refinement, the multi-scale paradigm can improve robustness.
> > >
> > > Table1 (Ablation results on VQ tokenizer)
> > > | Method          | Ref. | Ref.+ | Ref.g | Average | Time   |
> > > |-----------------|------|-------|-------|---------|--------|
> > > | Single-scale VQ | 82.1 | 71.8  | 65.8  | 73.23   | 5.50s  |
> > > | Multi-scale VQ  | 80.5 | 76.7  | 70.4  | 75.87   | 1.28s  |
> > >
> > >
> > > Thank you again for your recognition of our technical contributions and your suggestions for improvement.
> > >
> > > Best regards,
> > >
> > > The Authors
> > >
> > > > [1] Pan X, Shukla S N, Singh A, et al. Transfer between modalities with metaqueries[J]. arXiv preprint arXiv:2504.06256, 2025.
> > >
> > > > [2] Xie E, Chen J, Chen J, et al. Sana: Efficient high-resolution image synthesis with linear diffusion transformers[J]. arXiv preprint arXiv:2410.10629, 2024.
> > >
> > > > [3] Wu C, Chen X, Wu Z, et al. Janus: Decoupling visual encoding for unified multimodal understanding and generation[C]//Proceedings of the Computer Vision and Pattern Recognition Conference. 2025: 12966-12977.

---

### Official Review · Reviewer_pPbW · 2025-07-03

**Clarity:** 2
**Significance:** 2
**Originality:** 2
**Rating:** 3
**Confidence:** 5

**Summary:**

This paper introduces a novel approach to image segmentation by integrating it into a multimodal large language model (MLLM) framework using an auto-regressive generation strategy. The method aims to generate dense segmentation masks by treating them as visual tokens, which are decoded via a universal VQ-VAE detokenizer. To improve efficiency, the authors propose a next-scale prediction strategy that reduces inference latency. The method is evaluated across several segmentation benchmarks.

**Questions:**

Please see weakness points.

**Ethical Concerns:**

["NO or VERY MINOR ethics concerns only"]

**Final Justification:**

As noted in comments, the contribution of this work is overstated and falls short of making a meaningful step toward AGI. The proposed method appears as a combination of existing techniques without clear motivation or conceptual insight. Moreover, the experimental evaluation is insufficient to support the claimed benefits of integrating image generation and segmentation. Given this disconnect between the claims and the evidence, I maintain my original rating.

**Limitations:**

yes

**Paper Formatting Concerns:**

n.a.

**Quality:**

3

**Strengths And Weaknesses:**

## Strengths

- The paper addresses an emerging and important problem: how to unify dense prediction tasks like segmentation within the auto-regressive MLLM framework.

## Weaknesses

- Motivation is confusing
    - The overall motivation of the paper is unclear. While the title emphasizes segmentation with auto-regressive models, much of the method closely mirrors existing work (e.g., VAR), with little clarity on what differentiates this approach specifically for segmentation.
    - If the main contribution is the use of a universal detokenizer for both image generation and segmentation, the benefits of doing so are not convincingly argued. Segmentation masks are typically simpler than natural images, and prior work (e.g., HiMTok) has demonstrated efficient task-specific decoding. The universal approach seems less efficient, as shown in Table 6 where segmentation requires more tokens. The paper should explicitly clarify why a universal detokenizer is preferable and what advantages it offers.

- Details about segmentation part is unclear
    - Given that segmentation is the focus, the paper lacks insight into how this task is uniquely handled. The current presentation makes it appear as though the VAR framework is simply fine-tuned with segmentation data, which may not constitute a strong methodological contribution.
    - The integration of segmentation into the auto-regressive MLLM is not clearly described. Without this clarity, it is difficult to assess the novelty and effectiveness of the proposed method. This weakens the interpretability and credibility of the experimental results.

- Some experimental conclusions need further analysis. For example:
    - Some experimental findings, such as those in Table 5, are not thoroughly analyzed. For example, improvements may stem from dataset-specific artifacts (e.g., overly simplistic language annotations), rather than from the claimed benefits of multi-task training.
    - The paper does not adequately investigate whether the joint training of segmentation and image generation leads to mutual benefit. Further ablation or control experiments are needed to identify the source of performance gains.

## Conclusion

While the paper presents an interesting direction in leveraging auto-regressive MLLMs for segmentation, the contributions are currently underdeveloped. The motivation needs to be clarified, technical integration of segmentation should be described in more detail, and experimental claims require deeper analysis.

---

> ### Author Rebuttal · Authors · 2025-07-30
>
> Thank you very much for your positive feedback on our results. We realize that some aspects of our motivation and framework were misunderstood, so allow us to restate them clearly.
>
> **Motivation**
> - Our goal is to **unify dense perception tasks such as segmentation inside a single MLLM framework**, so that multimodal understanding, image generation, and visual perception coexist in one general-purpose framework. To this end, we **re-formulate segmentation as an autoregressive image-generation problem**.
>  - Our research demonstrates, for the first time, that dense perception can be integrated into a unified understanding-and-generation model and obtains more advanced performance than methods requiring additional  task-specific segmentation heads, offering a practical route toward unified AGI (L24–25).
>
> **Method**
> - We **extend the MLLM’s vocabulary with the VQ-VAE codebook indices**, enabling the MLLM to directly predict visual tokens for image generation (L144–147).
> - We adopt a **multi-scale VQ-VAE** to accelerate inference, **not** to be confused with VAR.
>    - VAR is a pure generative model with no multimodal understanding capability, whereas our framework **endows an existing MLLM with both image-generation and segmentation abilities**.
>
>
>
> ### **W1.1: Motivation of the paper**
>
> **A1.1:**
> We hope our earlier clarifications have resolved any confusion regarding the motivation of our work.
>
> **1. We aim to provide a concrete technical demonstration that dense perception can be unified with multimodal understanding and generation inside a single, end-to-end model.**
> - In our work, segmentation (a perception task) is accomplished through generation. This demonstrates a path where understanding and generation are not separate capabilities but are deeply intertwined. We hope this will inspire further research into more holistic, unified architectures.
>
> **2. Our approach is fundamentally different from VAR which is a pure generative model.**
> - VAR lacks image-understanding capabilities and cannot perform complex visual-perception tasks.
>   - In our framework, the MLLM is the core engine for both understanding and perception. We use image generation via a VQ-VAE simply as the expressive medium for the MLLM to output its perceptual results.
>
> **3. Rather than fine-tuning an existing unified understanding-and-generation model, we made unique design choices specifically to address segmentation-specific challenges.**
> - **Pixel-level accuracy.** Segmentation demands fine-grained precision beyond semantic alignment. Hence we let the MLLM **directly predict image tokens** instead of acting merely as a text encoder that feeds semantic embeddings to external decoders.
> - **Latency.** Perception tasks require fast response. Multi-scale VQ tokens allow **all tokens of the current scale to be generated in parallel within a single forward pass**, cutting per-sample latency from 5.50 s (single-scale) to 1.28 s.
> - **Understanding boosts segmentation.** To preserve the MLLM’s reasoning power—crucial for referring segmentation—we keep **continuous visual embeddings** as input features.
> ### **W1.2:  Advantages of using a universal tokenizer**
>
> **A1.2:**
> Thank you for highlighting the role of the universal tokenizer.
> Its use is directly tied to our motivation: we aim for a single, unified model, so we need a single, unified tokenization scheme that simultaneously enables image segmentation and unlocks image-generation capability.
>
> **1. Using a universal tokenizer enables understanding, segmentation, and generation tasks to be performed simultaneously within a unified MLLM framework.**
> - By avoiding task-specific segmentation heads and instead relying on a general-purpose generative paradigm, our framework can jointly perform multimodal understanding, visual perception, and image generation—offering a practical path toward general-purpose multimodal intelligence (L54–55).
>
> **2. Our multi-scale tokenizer achieves higher segmentation performance with less inference time, demonstrating superior efficiency.**
> - Table 6 shows that, despite producing more tokens, **our multi-scale universal tokenizer requires less inference time and delivers higher segmentation accuracy, confirming its efficiency.**
> - The key message of Table 6 is that multi-scale tokenization yields a clear inference-speed advantage, which is why we adopt it as our universal tokenizer while still gaining the ability to generate images.
>
> ### **W2.1: Insight into how segmentation task is uniquely handled**
>
>
> **A2.1**
> We hope our earlier explanations have resolved this concern; please let us know if any questions remain.
>
> **1. To meet the pixel-level accuracy and latency demands of segmentation, we deliberately designed the framework around direct visual-token prediction by the MLLM and multi-scale image generation.**
> - To the best of our knowledge, we are the first to use multi-scale VQ-VAE indices as the **native output of an MLLM** within a unified understanding-and-generation pipeline, which constitutes a **methodological contribution.**
>
> **2. Our architecture couples an MLLM with a multi-scale VQ-VAE decoder, endowing the same model with multimodal understanding, segmentation, and generation (Fig. 2).**
> - VAR, in contrast, is a pure image-generation transformer that
>   - (i) lacks multimodal understanding and
>   - (ii) can only ingest a single semantic embedding,
>   - (iii) making it **impossible to fine-tune for segmentation**.
> - We start from a general multimodal LLM, equip it with image-generation capability, and then fine-tune the entire model for segmentation using a **single cross-entropy objective**.
>
> ### **W2.2  How segmentation is integrated into the auto-regressive MLLM**
>
> **A2.2**
> As illustrated in Figure 2, **our model directly employs the MLLM itself for image generation**, and we re-cast segmentation as an image-generation task. In this way, multimodal understanding, segmentation, and image generation are all handled by the same autoregressive MLLM without any external segmentation heads.
>
> To be explicit:
> - Our research question is how to enable an MLLM—originally designed for understanding—to perform both image generation and segmentation, rather than how to use a standalone generative model for segmentation.
> - If any part of this integration remains unclear, we warmly welcome further discussion and are happy to clarify any details.
>
> ### **W3.1 Further analysis experimental conclusions**
>
> **A3.1:**
> We would like to first emphasize that:
>
> **1. The "multi-task" in Table 5 is obtained by jointly fine-tuning the MLLM on multimodal understanding data and segmentation data only; no image-generation data are involved.**
> - Image-generation data are used solely in the Function-Extension stage to demonstrate that the same unified framework can also generate images.
>
> **2. In the paper L255-257, we analyzed the performance improvement from the model's understanding ability from both absolute and relative perspectives.**
> - Absolute gains: Simply adding understanding data yields higher cIoU on all three referring-segmentation benchmarks, confirming that stronger language understanding directly benefits segmentation.
> - Relative gains: RefCOCO+ and RefCOCOg, which contain more semantically challenging referring expressions, enjoy the largest relative improvements.
>
> **3. From another perspective, even considering only the complexity of the language annotations, it is logical that training on a richer and more diverse set of linguistic instructions (from the understanding tasks) would enhance the model's ability to follow the complex commands required for accurate segmentation.**
>
>
> ### **W3.2 The joint training of segmentation and image generation**
>
> Thank you for this insightful comment.
>
> **1. In the present work we focus on the mutual reinforcement between multimodal understanding and segmentation**
> - Table 4 demonstrates that segmentation training, in turn, benefits visual grounding and reduces hallucination.
> - Table 5 shows that stronger understanding improves segmentation, especially on reasoning-heavy benchmarks.
>
> **2. A systematic study of the interaction between image generation and segmentation is beyond the current scope and is left as future work.**

---

### Official Review · Reviewer_5AFh · 2025-07-03

**Clarity:** 3
**Significance:** 2
**Originality:** 2
**Rating:** 4
**Confidence:** 3

**Summary:**

This work presents a new model for language-prompted image segmentation. The proposed method employs a pre-trained MLLM model (e.g., InternVL) as the backbone architecture, equipped with newly trained VQ tokens that encodes the segmentation mask to the MLLM latent space, therefore the model can be trained to predict the VQ tokens corresponding to the target mask. A special design of this VQ tokenization design is employing the multi-scale structure (up to k=10 different scales), where the MLLM can iteratively refer to the previously decoded tokens at larger scales (i.e., smaller k's), and produce next tokens at smaller scales (i.e., larger k's).

In the experiments, the proposed model is compared with several image segmentation models that also utilizes MLLM, and the ablation study is performed on different training strategies of the proposed method.

**Questions:**

- Please provide more detailed analyses on the VQ tokenization of the masks. How significant is it to train the proposed multi-scale VQ tokens for the segmentation masks? Why can it better perform than more conventional designs to handle masks, such non-discrete features?

- How does the proposed method generalize when other MLLMs are employed? For example, could we reproduce a similar results in llama MLLM architecture?

**Ethical Concerns:**

["NO or VERY MINOR ethics concerns only"]

**Final Justification:**

I appreciate the authors' comprehensive response to my review. After reviewing the additional experimental results, I find the paper's findings more convincing. As the authors themselves acknowledged, the model's reasoning ability warrants further attention. I encourage them to include their analyses covering different MLLM backbones in the final manuscript.

In summary, I am updating my score to Borderline accept. I believe this method could be a valuable addition to the research community.

**Limitations:**

Despite the use of MLLM backbone, the reasoning ability of the proposed model is not well discussed. Since the training data is the mixture of the conventional image segmentation dataset and the multimodal understanding (without the notion of masks), the model might not retain reasoning ability in terms of the image segmentation tasks.

**Quality:**

2

**Strengths And Weaknesses:**

- Strength of the paper includes easy-to-follow presentation and straightforward model design

- Missing crucial ablation study of the proposed modules, for example, the impact of multi-scale VQ tokenization, compared to the basic single-scale VQ tokenization, or non-VQ mask encoding/decoding methods.

- The method relies on +20 different large-scale datasets for image segmentation, generation, and multi-modal understanding tasks. However, the specific details on the training objective design to handle each dataset and task is insufficiently discussed. Furthermore, as the training dataset of the baseline methods are not clearly compared, it is hard to fully understand whether the improved performances of the proposed method is mainly due to larger training data, or the proposed key design ideas.

---

> ### Author Rebuttal · Authors · 2025-07-30
>
> Thank you very much for your valuable feedback. Please allow us to first clarify your two key confusions:
>
> 1. **We strictly use subsets of the data employed by the previous SOTA method, HiMTok [2], to ensure a fair comparison.**
> - Specifically, we train on 402 K segmentation samples versus HiMTok’s 2.91 M, and 1.25 M multimodal understanding samples versus HiMTok’s 4.2 M.
>
> 2. **Our method can be seamlessly extended to any MLLM framework, and all tasks are trained using a unified cross-entropy loss objective.**
> - All tasks—segmentation, visual understanding, and generation—are trained under the same autoregressive paradigm and optimized solely with cross-entropy loss (Fig. 2 and L168–170).
>
> ### **W1 & Q1: Ablation study and more detailed analyses on the VQ tokenization of the masks.**
>
> **A1:**
> Thank you very much for this insightful suggestion. During the rebuttal period we conducted an ablation study in which we replaced our multi-scale VQ tokenizer with the single-scale VQ-GAN pretrained in Janus [1] and trained it on the same pure-segmentation data. We will include a detailed discussion of these results in the final version of our paper.
>
> Our choice to use multi-scale VQ tokens for training is motivated by the unique challenges posed by the image segmentation task:
>
> 1. **We use multi-scale VQ instead of single-scale to meet the latency requirements of perception tasks.**
> - Multi-scale tokens allow the MLLM to predict all visual tokens of the current scale in parallel, dramatically reducing inference time. Under identical hardware and framework settings, the single-scale variant (token-by-token generation) needs **5.50 s per sample**, whereas our **multi-scale design finishes in 1.28 s** (Table 1).
> 2. **We enable the MLLM to directly output VQ image tokens, rather than semantic embeddings, to meet the pixel-level precision requirement of segmentation.**
> - Segmentation demands higher low-level and pixel-wise accuracy than generic image generation. We therefore let the MLLM directly predict image tokens instead of producing semantic embeddings for an external decoder. When we instead fed MLLM-generated condition embeddings to a DiT decoder, **the resulting masks suffered from severe pixel inconsistency.**
>
> For ablation with single-scale VQ tokenization, we used the VQGAN pretrained by Janus as the single-scale VQ tokenizer, and trained it solely with segmentation data. We report the segmentation metric "giou," which represents the per-sample IoU averaged over the dataset.
> - As show in Table 1, the multi-scale scheme not only yields a clear speed advantage but also improves robustness via its inherent coarse-to-fine refinement, as visualized in Fig. 3.
>
> Table1 (Ablation results on VQ tokenizer)
> | Method          | Ref. | Ref.+ | Ref.g | Average | Time   |
> |-----------------|------|-------|-------|---------|--------|
> | Single-scale VQ | 82.1 | 71.8  | 65.8  | 73.23   | 5.50s  |
> | Multi-scale VQ  | 80.5 | 76.7  | 70.4  | 75.87   | 1.28s  |
>
> Compared with non-VQ pipelines that let the MLLM act only as a text encoder and rely on semantic-similarity decoding (L82), our approach performs true pixel-level understanding and significantly outperforms such methods (Table 2 in paper). Moreover, those pipelines break the unification of understanding and generation that our framework is designed to achieve.
>
> ### **W2: training objective and datasets scale compare to baseline**
>
> **A2:**
> We hope our earlier replies have already clarified this point. To summarize succinctly:
> 1. **We use an unified objective.**
> - Across all tasks—segmentation, multimodal understanding, and generation—we supervise the model with **cross-entropy loss** under one unified paradigm.
>   - Image-generation data are employed **only** in the optional Function-Extension stage.
>
> 2. **Our data scale is significantly smaller than the compared method.**
> - Our method achieves higher segmentation performance while using significantly less data compared to the previous SOTA method, HiMTok [2] **(402K vs. 2.91M samples).**
>    - This further highlights the effectiveness of our proposed approach. For a direct comparison, the datasets used by HiMTok [2] are listed in Table 1 of their paper.
>
>
> ### **Q2: How does the proposed method generalize when other MLLMs are employed?**
>
> **A3:**
> Thank you for this valuable question. We will include the following ablation in the final version.
>
> **1. Our method enables image segmentation in MLLM frameworks by incorporating the VQVAE codebook into the MLLM's token space, allowing seamless extension to any MLLM architecture.**
> - In practice, one important detail is that during image generation, the visual tokens predicted by the MLLM are not directly fed as input for the next prediction step. Instead, the predicted image token IDs are mapped to their corresponding visual features in the VQVAE codebook, upsampled to the next scale, and then passed through a Generation Adapter before being provided as input to the MLLM.
>
> **2. We conducted extended experiments using LLaVA-1.5, which is an MLLM framework based on LLaMA-2.** The experimental results show in Table 2.
> - Our pipeline successfully endows the LLaMA-based LLaVA-1.5 with segmentation capability.
> - As noted in our ablation (L255–256), reasoning-segmentation performance is strongly correlated with the MLLM’s understanding ability.
> While LLaVA-1.5’s understanding capability is notably weaker than InternVL2.5 (with InternVL2.5 averaging ~68.1 and LLaVA-1.5 ~46 as shown in InternVL2.5’s Figure 1), segmentation results with LLaVA-1.5 are accordingly lower.
> Nevertheless, **when the same backbone is used, our method still outperforms HiMTok,** confirming that the gains are not merely due to a stronger backbone.
>
> Table 2 (Extension experiment with LLaVA-1.5 (LLaMA-2 backbone))
> | Method          | backbone    | Ref. | Ref.+ | Ref.g |
> |-----------------|-------------|------|-------|-------|
> | HiMTok          | InternVL2.5 | 81.1 | 77.1  | 75.8  |
> | ARGenSeg-llava  | LLaVA–1.5   | 72.7 | 68.3  | 69.1  |
> | ARGenSeg-internvl | InternVL2.5 | 82.2 | 77.9  | 78.4  |
>
> ### **Limitation:  the model might not retain reasoning ability in terms of the image segmentation tasks.**
>
> **A4:**
> Thank you for raising this important concern. We shared the same worry and therefore ran targeted experiments to verify whether adding segmentation data would cause the model to be unable to maintain its reasoning ability. **Our findings show that it does not.**
>
> **1. Theoretical justification**
> - During both multimodal reasoning and segmentation training, the model **never treat the mask as input**; the mask is used only as a supervision signal for the perceptual task.
> - In our framework, multimodal understanding and image generation are decoupled at the output stage only: text tokens are decoded by the text tokenizer, while image tokens are decoded by the VQ-VAE decoder. There is no architectural conflict that would degrade reasoning.
>
> **2. Empirical verification**
> - In response to your suggestion, we have further evaluated the model’s reasoning abilities from multiple perspectives. Our experimental results in Table 3 show that adding segmentation data does not cause the model to lose its reasoning capability.
> - Although the absolute scores drop slightly on some benchmarks, we attribute these small decreases not to the segmentation task itself, but to the **much smaller and lower-quality understanding corpus** we used (1.25 M vs. InternVL2.5’s full 16.3 M).
> - To confirm this hypothesis, we fine-tuned InternVL2.5 on understanding data only for an increasing number of steps; performance declined monotonically, mirroring the trend we observe when segmentation data are added.
>   - As explained in L194–195, InternVL’s open-source release only covers version 1.2, limiting the data we can legally use.
>
> Table3 (Ablation experiment of unifying MLLM model reasoning ability)
> | Method          | POPE   | TextVQA |VQAv2 |MMMU-val | AI2D |
> |-----------------|--------|---------|---------|---------|---------|
> | InternVL2.5-ft–1ep | 86.73 | 63.54  |80.40 | 43.7  | 78.7  |
> | InternVL2.5-ft–4ep | 86.01 | 59.73  | 79.28 | 36.8 | 74.8  |
> | ARGenSeg        | 87.57 | 56.98  | 77.87 | 33.4 | 69.6 |
>
> [1] Wu C, Chen X, Wu Z, et al. Janus: Decoupling visual encoding for unified multimodal understanding and generation[C]//Proceedings of the Computer Vision and Pattern Recognition Conference. 2025: 12966-12977.
>
> [2] Wang T, Cheng C, Wang L, et al. Himtok: Learning hierarchical mask tokens for image segmentation with large multimodal model[J]. arXiv preprint arXiv:2503.13026, 2025.

---

> > ### Comment · Reviewer_5AFh · 2025-08-06
> >
> > I appreciate the authors' comprehensive response to my review. After reviewing the additional experimental results, I find the paper's findings more convincing. As the authors themselves acknowledged, the model's reasoning ability warrants further attention. I encourage them to include their analyses covering different MLLM backbones in the final manuscript.
> >
> > In summary, I am updating my score to Borderline accept. I believe this method could be a valuable addition to the research community.

---

> > > ### Author Response · Authors · 2025-08-07
> > >
> > > Dear Reviewer 5AFh,
> > >
> > > We are glad that your concerns have been addressed, and we greatly appreciate your recognition of our work as a valuable addition to the research community.
> > >
> > > Thank you again for your constructive and insightful suggestions. We will incorporate the content you recommended into the final version of our paper.
> > >
> > > Best regards,
> > >
> > > The Authors

---

### Official Review · Reviewer_6HDm · 2025-07-03

**Clarity:** 3
**Significance:** 2
**Originality:** 3
**Rating:** 4
**Confidence:** 2

**Summary:**

The authors propose a unified multimodal understanding and generation model by training MLLMs to predict image tokens and decode into predicted masks. To speed up inference, the authors integrate next-scale prediction, which aligns itself to the segmentation task. Qualitative evaluation shows improved performance on referring segmentation, while allowing down-stream finetuning to interactive segmentation and image generation via small amount of supervised data.

**Questions:**

My main concerns as below and please refer to the weakness section for more details.
- Method: What insight can be drawn from the proposed method? How and why are certain design choices (e.g., directly predicting image latent, adding <gen_start> and <gen_end>, etc) made?
- Evaluation: How significant does this method improve over baseline? What are some particular scenarios where the proposed method consistently outperform baseline?

As I do not work directly on this topic, I am happy to change my recommendation as I await to read other reviewer’s comments and the author's response to my questions.

**Ethical Concerns:**

["NO or VERY MINOR ethics concerns only"]

**Final Justification:**

I would like to first thank the authors for their efforts in the new response, as many of my concerns are addressed. While I still find the technical novelty of the proposed method to be lacking, I find the paper to be valuable with enough scope to the broader research community. Therefore, I am inclined to keep my recommendation as borderline accept.

**Limitations:**

Limitations are briefly discussed in the Appendix, though not particularly regarding the proposed method.

**Quality:**

3

**Strengths And Weaknesses:**

### Strengths
- Integrating dense perception tasks with generative models is an important research direction.
- The work is well-motivated: tasks outputting dense predictions like image segmentation are inherently difficult to process along natural language instructions, making a unified framework an intuitive investigation.
- The idea of localization first and details second (native for segmentation) fits the auto-regressive scale prediction paradigm well.
- Quantitate evaluation shows improved performance over baseline methods. Though I am less familiar with past works and cannot judge the completeness of the evaluation benchmarks.
- Nice extensions to interactive segmentation and image generation via down-stream finetuning. This further verifies the capabilities of the proposed method.

### Weakness
- The proposed method, although straight-forward, has limited technical novelty and appears more engineering and similar to existing works (e.g., Janus and VARGPT). Though I am less familiar with the existing literature and not best fit to judge the architectural novelty.
- There is somewhat limited discussions and intuitions in the method section. For instance, the Section 3.2 and 3.3 contain mostly descriptions on the design choice with limited to no insights.
	- Discussions on why/how this particular architecture is chosen may improve the scope of the work.
- No statistical significance is reported. Given how similar some of the performances are in Tab. 2, having statistical significance is important to measure performance improvement.
- Limited qualitative results in the manuscript &Appendix. In my opinion, the benefit of MLLM comes from its ability to handle inputs in the wild / edge cases and should be highlighted via more examples.

---

> ### Author Rebuttal · Authors · 2025-07-30
>
> Thank you very much for your positive feedback on our work. We apologize for any confusion caused by our writing, and we hope that the following responses will address your concerns.
>
> To help clarify the architectural novelty of our approach, please allow us to first address your second concern regarding the insights and design of our method.
>
> ### **W2: insights and design choices**
>
> **A2:** Thank you for this valuable suggestion. We will add a clearer, more detailed discussion of all design choices and the insights behind them in the final version.
>
> Below we summarize the key insights that guided our design decisions when integrating segmentation into a unified MLLM framework, and then explain the corresponding architectural choices:
> 1. **Insight1:** The strong **comprehension capabilities** of MLLMs can significantly boost segmentation performance, especially for reasoning-intensive tasks.
> - *Design1:* We therefore cast segmentation as an image-generation problem and **embed it into the unified MLLM framework.** (See ablation study, L255–258.)
>
> 2. **Insight2:** Unlike general image generation—which primarily emphasizes semantic alignment—segmentation demands **pixel-level accuracy.**
> - *Design2:* Instead of producing semantic embeddings that are later decoded by a downstream model, we let the MLLM **directly predict image tokens.** (See Introduction and Related Work, L42–44 and L83–84.)
>
> 3. **Insight3:** Segmentation is a perception task that often requires **faster response times.**
> - *Design3*: We adopt a **multi-scale VQ-VAE tokenizer** that allows all required visual tokens to be predicted in parallel during a single MLLM inference pass. (See Introduction and Table 6, L45–50 and L83–84.)
> 4. **Insight4:** We aim to incorporate dense perception tasks into a **unified AGI framework** that already supports multimodal understanding and generation.
> - *Design4:* By solving segmentation via autoregressive image generation, we demonstrate that **dense perception can be seamlessly integrated without specialized segmentation heads,** achieving higher performance and offering a practical path toward unified AGI models. (See Introduction, L24–25.)
>
> In addition to the points already raised, the following design decisions and analyses may also be helpful:
> - Continuous visual features for input image.
> To preserve the MLLM’s strong understanding capability (Insight 1), we keep the input visual features continuous. Discrete tokens inevitably degrade understanding performance, which is critical for reasoning-heavy segmentation tasks (see L36–37).
> - Single-stage fine-tuning for efficiency.
> We adopt a single-stage fine-tuning strategy. Our ablation study (L264–269) shows that adding a separate pre-training stage yields only marginal gains, so we omit it to improve training efficiency.
>
> Please let us know if any questions remain.
>
> ### **W1: technical novelty and differences from Janus and VARGPT**
>
> **A1:** We apologize for any confusion regarding the technical novelty of our work. First, we would like to emphasize that our main motivation is to explore the possibility of unifying dense perceptual tasks within the unified framework of understanding and generation.
> - We successfully demonstrate that an image-generation formulation can indeed solve segmentation and achieve SOTA results, offering a practical route toward unified AGI models. This finding alone carries intrinsic technical value and novelty.
>
> With respect to the architecture, our design differs fundamentally from both Janus and VARGPT, and these differences are driven by the unique challenges of segmentation:
> 1. **Janus vs ours: Janus relies on next-token prediction; in contrast, we adopt next-scale prediction.**
> -  This design originated from *latency challenge for perception tasks*. This allows all tokens of a given scale to be generated **in parallel** within a single forward pass, yielding significantly faster inference.
>    - Under identical settings, replacing our tokenizer with Janus’s VQ-GAN raises per-sample latency from **1.28 s to 5.50 s.**
> 2. **VARGPT vs ours: VARGPT appends an extra VAR transformer, while our approach enables the MLLM to directly output visual tokens.**
> - This is designed to tackle the *pixel-level accuracy and consistency challenge.* Unlike VARGPT, which has the MLLM **provide only semantic conditions to the generation head**, our model **directly predicts image tokens**, ensuring better pixel-level precision.
>   - When we experimented with letting the MLLM output semantic embeddings to a DiT decoder, the resulting masks roughly located the target but exhibited noticeable dilation or misalignment—clear evidence of **pixel-accuracy issues.**
>   - The technical report for VARGPT-v1.1 [1] only discusses image editing experiments related to style transfer. The visual results in Figure 9 reveal clear consistency issues, which we suspect stem from the same limitation.
>
> **To the best of our knowledge, we are the first to use multi-scale VQ-VAE visual indices as the MLLM’s native output within a unified understanding-and-generation framework.**
>
> ### **W3: statistical significance  to measure performance improvement.**
>
> **A3:** Thank you for raising this important point. We followed the evaluation protocol commonly adopted in the community and therefore did not include formal statistical-significance tests. Below we provide additional evidence that we hope will alleviate any concerns about the reliability of the reported improvements.
> - **Controlled training data.**
> To ensure a fair comparison, we deliberately used a strict subset of the training data employed by HiMTok—402 K image–segmentation samples versus their 2.91 M (L193–194). This rules out any advantage from additional data.
> - **Training stability.**
> Due to computational cost we did not repeat the main training runs. However, in our ablation study we re-initialized and re-trained the Gen projector multiple times; the resulting performance matched the original checkpoint within 0.2 cIoU, indicating low run-to-run variance.
> - **Deterministic inference.**
> Segmentation results are deterministic under our decoding strategy. We verified this by running inference (i) multiple times on the same checkpoint, (ii) on both single-GPU and multi-GPU setups, and (iii) with both our own re-implementation and the official evaluation script of HiMTok. All runs produced identical cIoU scores. Re-evaluating HiMTok with its official code also reproduced the numbers reported in their paper.
> - **Cross-dataset consistency.**
> Our method yields consistent gains across all benchmark datasets (Table 2). The aggregated metrics further corroborate that the improvements are not artifacts of a particular data split or sampling bias.
>
> Table1 (Average performance on all datasets)
> | Method           | Average   |
> |------------------|-----------|
> | LMMHiMTok-8B     | 77.675    |
> | ARGenSeg         | **79.6625**   |
> | LMMHiMTok-8B(ft) | 81.5875   |
> | ARGenSeg (ft)    | **83.35**     |
>
> ### **W4: Scenarios where our method outperforms the baseline.**
>
> **A4:**
> Thank you very much for your suggestion. Your point is both correct and valuable, and we will include more qualitative results in the final version of the paper.
>
> Following your suggestion, we exhaustively evaluated both HiMTok and ARGenSeg on the test set and visualized cases where ARGenSeg succeeds while HiMTok fails. These visualizations reveal two clear advantages of our coarse-to-fine, multi-scale generation scheme:
> 1. **Coarse localization of segmentation objects helps identify the target in multi-object scenarios:**
> Our approach allows for the rough localization of target objects, making it easier to identify the correct segmentation target when multiple similar objects are present in the image.
> - For example, when there are 6 bowls in an image, HiMTok may segment an entirely unrelated object for a simple instruction such as "bowl middle three red dots," while our method can accurately segment the intended target.
> - Other examples include: With 14 hot dogs, the instruction "left hand row of hot dogs fifth from the front" and, with 7 people present, "kid in black arms out." In such scenarios, our method is able to localize the correct target more successfully.
> 2. **Progressive refinement with multi-scale generation helps enhance mask quality:**
> The progressive, multi-scale refinement process enables our model to produce higher quality masks once the segmentation target is identified.
> - For instance, in the case of a teddy bear being held, both HiMTok and our method's coarse localization stage only identify a single part separated by an arm. However, during our method’s subsequent fine-grained refinement, the entire teddy bear is correctly segmented, with the arm successfully excluded from the final mask.
>
> ### **Q1: Method: insights and design choice**
>
> **A5:**
> We hope our responses **A2** have fully clarified the motivations and design choices behind our method. Should any questions remain, please do not hesitate to let us know—we are happy to provide further details.
>
> ### **Q2: Evaluation:  What are some particular scenarios where the proposed method consistently outperform baseline?**
>
> **A6:**
> Please refer to **A4.**
>
> [1] Zhuang X, Xie Y, Deng Y, et al. Vargpt-v1. 1: Improve visual autoregressive large unified model via iterative instruction tuning and reinforcement learning[J]. arXiv preprint arXiv:2504.02949, 2025.

---

> > ### Comment · Reviewer_6HDm · 2025-08-06
> >
> > I appreciate the authors’ efforts in addressing all of the concerns raised. After carefully reading over the feedback from other reviewers and the corresponding rebuttals, some of my concerns are addressed and here are some of my remaining comments:
> > - W1: I don’t find the difference between next-token prediction and next-scale prediction to be minor and not fully sufficient to show the technical novelty of the proposed method.
> > - W2: While I find the listed insights and design to be intuitive, I find them to be more-or-less straight-forward and have limited scope in adjacent fields. Furthermore, I agree with Reviewer TNWY that the current manuscript has limited segmentation-specific designs and insights discussed. Unified segmentation fades out during the method section.
> > - W3: While I understand the arguments made by the authors, I maintain my belief that statistical significance is important, especially given the performance gain in Tab. 2.

---

> > > ### Author Response · Authors · 2025-08-07
> > >
> > > Dear Reviewer 6HDm,
> > >
> > > Thank you very much for your careful reading of our responses and your thoughtful follow-up. We are glad that some of your concerns have been resolved. We highly appreciate your critical feedback and the time you invested in carefully reviewing our work.
> > >
> > > **Regarding W1:**
> > >
> > > First, we would like to clarify two points:
> > >
> > > **1. Our proposed coarse-to-fine progressive image segmentation is realized through the multi-scale image generation paradigm we introduced, rather than relying on the autoregressive prediction scheme.**
> > >
> > > **2. Next-token prediction corresponds to the single-scale image generation paradigm, and applying it directly to the segmentation task leads to reduced robustness.**
> > >
> > > Based on the above, we believe that the difference between next-token prediction and next-scale prediction is clear and significant.
> > >
> > > Additionally, we directly used the VQGAN pretrained by Janus as our VQ tokenizer for ablation study, and we report the segmentation metric "giou," which represents the per-sample IoU averaged over the dataset.
> > >
> > > - As shown in Table 1, the multi-scale image generation paradigm provides a significant speed advantage (1.28s vs 5.50s).
> > > - The coarse-to-fine progressive paradigm also improves the robustness of image segmentation.
> > >
> > > Table1 (Ablation results on VQ tokenizer)
> > > | Method          | Prediction |Ref. | Ref.+ | Ref.g | Average | Time   |
> > > |------------|-----|------|-------|-------|---------|--------|
> > > | Single-scale VQ | next-token | 82.1 | 71.8  | 65.8  | 73.23   | 5.50s  |
> > > | Multi-scale VQ  | next-scale | 80.5 | 76.7  | 70.4  | 75.87   | 1.28s  |
> > >
> > > **Regarding W2:**
> > >
> > > Thank you for your suggestion. We will enhance the discussion of segmentation-specific insights and designs in the technical part of our final version, and illustrate them with concrete examples. Please refer to our latest response to Reviewer TNWY for more details.
> > >
> > > Additionally, we believe that our insights can offer inspiration for adjacent fields:
> > > - Our insight into the importance of pixel-level accuracy in image generation may benefit future research on scene consistency in domains such as image editing.
> > > - Our efforts to improve inference efficiency (e.g., by eliminating dedicated image decoders beyond what VARGPT adopted) could further promote real-time capabilities in fields like embodied intelligence.
> > >
> > > **Regarding W3:**
> > >
> > > Thank you again for your suggestion. Although our computational resources are limited, we will try our best to conduct multiple trainings and include statistical significance analysis in our camera-ready version.
> > >
> > > Thank you again for your positive evaluation of our work. We will incorporate your suggestions into the final version of our manuscript.
> > >
> > > Best regards,
> > >
> > > The Authors

---

### Comment · Area_Chair_r2En · 2025-08-04

Dear Reviewers,

If you have not done so already, please review the authors' rebuttal to your comments and the other reviews.  Please submit any further questions for the authors promptly to allow time for discussion.

Please also remember to update your ratings and final justification if your concerns have been addressed. If ratings are not updated, clearly explain any remaining concerns in your final justification.

As a reminder, the author-reviewer discussion period ends on August 6th, 11:59 PM AoE.

Best, Your AC

---

### Note · Authors · 2025-08-16

We sincerely thank all reviewers for their time, valuable feedback, and constructive suggestions, as well as the AC, SACs, and PCs for organizing and coordinating the review process.

Our ARGenSeg integrates image segmentation into a unified MLLM framework via an autoregressive generation paradigm. To address the unique challenges of segmentation, we designed the framework so that the MLLM directly outputs image tokens for pixel-level accuracy and utilizes multi-scale image generation for high responsiveness and robustness through coarse-to-fine refinement. Our experiment results are the first to show that unified MLLM models can perform state-of-the-art segmentation without any extra task-specific segmentation heads, providing an effective technical pathway for unified AGI.

We are encouraged that the significance and contribution of our work were acknowledged by all reviewers:
- 6HDm: "Integrating dense perception tasks with generative models is an important research direction."
- pPbW: "The paper addresses an emerging and important problem."
- 5AFh: "I believe this method could be a valuable addition to the research community."
- TNWY: "the paper has technical contributions."

During the rebuttal, we made every effort to address all concerns.
For 6HDm and TNWY's novelty questions, we clarified our dedicated unified MLLM design for segmentation.
To 5AFh’s concerns on experimental reliability, we clarified that our training data are strict subsets of baseline data to ensure a fair comparison.
For pPbW’s misunderstandings, we provided detailed responses to all questions to clarify our motivation and method.

Following the suggestions of 6HDm, 5AFh, and TNWY, we performed extensive additional experiments, including: qualitative coarse-to-fine segmentation results, ablation with single-scale generation, generalization to other backbones, reasoning ability verification, quantitative results for interactive segmentation, and high-resolution segmentation feasibility. These experiments aided reviewer understanding, and led 5AFh to state, "After reviewing the additional experimental results, I find the paper's findings more convincing," and to raise his score.

All new experiments and discussions from the rebuttal will be included in the final version of ARGenSeg, and we will further strengthen the technical parts with concrete examples. Once again, we thank all reviewers for their feedback, which has greatly improved the quality and clarity of our work.

---

### Decision · Program_Chairs · 2025-09-17

**Decision:**

Accept (poster)

**Comment:**

This paper proposes ARGenSeg which integrates image segmentation into a unified MLLM framework via autoregressive generation.  The final ratings are 3 borderline accepts and 1 borderline reject.  The reviewers are overall positive about the importance of the problem and technical contribution.  Concerns regarding novelty, experimental reliability, and the need for additional experiments (qualitative results, ablation with single-scale generation, other backbones, interactive segmentation, and high-resolution segmentation) were largely addressed by the rebuttal and discussion. Concerns remained regarding overstated contributions and limited insights.  This is a borderline paper.   The AC carefully reviewed the paper, reviews, rebuttal and discussion, and deem that the positives of the overall framework and promising results outweigh the negatives, and hence recommend acceptance.  Please incorporate the additions during the rebuttal/discussion phase into the final version.